# 3D Occlusal Tooth Wear Assessment in Presence of Limited Changes in Non-Occlusal Surfaces

**DOI:** 10.3390/diagnostics11061033

**Published:** 2021-06-04

**Authors:** Nikolaos Gkantidis, Konstantinos Dritsas, Christos Katsaros, Demetrios Halazonetis, Yijin Ren

**Affiliations:** 1Department of Orthodontics and Dentofacial Orthopedics, University of Bern, CH-3010 Bern, Switzerland; dritsaskonstantinos@gmail.com (K.D.); christos.katsaros@zmk.unibe.ch (C.K.); 2Department of Orthodontics, W.J. Kolff Institute, University Medical Center Groningen, University of Groningen, 9700RB Groningen, The Netherlands; y.ren@umcg.nl; 3Department of Orthodontics, School of Dentistry, National and Kapodistrian University of Athens, GR-11527 Athens, Greece; dhal@dhal.com

**Keywords:** tooth wear, measurement method, three-dimensional imaging, surface model, 3D superimposition, quantitative assessment

## Abstract

The study aimed to develop an accurate and convenient 3D occlusal tooth wear assessment technique, applicable when surfaces other than the occlusal undergo changes during the observation period. Various degrees of occlusal tooth wear were simulated in vitro on 18 molar and 18 premolar plaster teeth. Additionally, their buccal and lingual surfaces were gently grinded to induce superficial changes and digital dental models were generated. The grinded and the original tooth crowns were superimposed using six different 3D techniques (two reference areas with varying settings; gold standard: GS). Superimposition on intact structures provided the GS measurements. Tooth wear volume comprised the primary outcome measure. All techniques differed significantly to each other in their accuracy (*p* < 0.001). The technique of choice (CCD: complete crown with 30% estimated overlap of meshes) showed excellent agreement with the GS technique (median difference: 0.045, max: 0.219 mm^3^), no systematic error and sufficient reproducibility (max difference < 0.040 mm^3^). Tooth type, tooth alignment in the dental arches, and amount of tooth wear did not significantly affect the results of the CCD technique (*p* > 0.01). The suggested occlusal tooth wear assessment technique is straightforward and offers accurate outcomes when limited morphological changes occur on surfaces other than the occlusal.

## 1. Introduction

Tooth wear is the loss of tooth structure that occurs during time; it has very high occurrence in the population. It is of multifactorial etiology and it can be a result of normal function, parafunction, or of environmental factors, such as highly acidic food. Apart from dental esthetic impairment, excessive tooth wear may even alter facial morphology or speech, negatively impacting quality of life. In contemporary societies, tooth wear is considered an important problem since the increased human life span and patient demands led to the need for presence of natural teeth in the dentition for several decades [1]. Progress in dental sciences has facilitated this purpose and also offered various opportunities for prevention, as well as restoration, of tooth wear [2,3].

Proper and timely diagnosis, through accurate monitoring of tooth wear progression and treatment outcome assessment, is the first step for successful tooth wear management [2,4]. Therefore, several previous studies have addressed this issue and suggested various qualitative and quantitative approaches to assess tooth wear. The qualitative approaches, such as the Eccles index, the Tooth Wear Index, the Lussi index, or the Basic Erosive Wear Examination, are commonly used, since they are relatively simple, they do not require special equipment, and thus, they can be easily incorporated in large-scale studies or used for clinical screening. However, apart from the specific strengths and limitations of each individual index, the main shortcoming of qualitative approaches is the reduced accuracy, including reproducibility [5,6]. On the other hand, there are highly reproducible quantitative techniques, such as digital profilometry and 3D surface matching techniques, but their trueness was, in most cases, not assessed due to the absence of a gold standard reference (true value) [7]. Furthermore, certain methods are too complicated to be applied in clinical practice, because they require special equipment, as well as impression taking and construction of physical models. This increases costs and sources of error and reduces the applicability of these methods [7,8,9]. For any quantitative method that uses digital models at any stage, advances in imaging techniques over time facilitate its applicability through the enhancement of accuracy and the reduction of costs.

In previous studies, we suggested accurate 3D superimposition methods to assess tooth wear on intraorally obtained serial digital dental models, using techniques applicable under various clinical conditions [10,11]. Due to the rapid incorporation of intraoral scanners in contemporary dentistry [12,13], these approaches can be easily applied in clinical settings, through the use of relevant software. One previous study tested wear assessment in teeth where all crown surfaces, apart from the worn occlusal aspects, remained intact over time [10]. Another previous study tested the scenario of extensive changes over time, on non-occlusal surfaces, additionally to the occlusal tooth wear. Such changes were simulated by the placement of a wire retainer on the lingual aspects of the tested teeth [11]. However, these conditions do not cover the whole spectrum of possible clinical occurrences. Tooth crown morphology might also be subjected to limited changes in non-occlusal surfaces, by dental caries, fillings, tooth wear, or other factors [14,15,16]. Such conditions may complicate the superimposition procedure and affect the outcomes. Thus, in the present study, the in vitro methodology that was previously developed, validated and applied to assess tooth wear under different clinical scenarios [10,11], was used to define a 3D superimposition method that can accurately measure occlusal tooth wear on teeth that undergo limited changes in non-occlusal crown surfaces, additional to the occlusal wear.

## 2. Materials and Methods

### 2.1. Sample

The study sample comprised 16 dental plaster models (type IV plaster, white color, Fujirock EP Premium, GC, Leuven, Belgium) depicting dental arches of both jaws, with varying alignment status. More specifically, eight models with crowding ≤ 1 mm (four maxillary and four mandibular) and an additional eight models with crowding ranging from 4 to 10 mm (four maxillary and four mandibular) were retrieved from the archive of the Department of Orthodontics and Dentofacial Orthopedics, University of Bern, Switzerland. The eligibility criteria included the presence of all permanent teeth till the second molars and no extreme morphological variation, assessed through visual inspection. More details on sample composition and characteristics were reported previously [10].

### 2.2. Tooth Wear Simulation

According to a modified previously published protocol [10,11], 18 premolars and 18 molars, equally distributed among the dental models, comprised the testing material. On these, various degrees of tooth wear, in terms of vertical reduction (approximately 0.5, 1, and 2 mm), were simulated through manual grinding of the occlusal tooth surfaces [10,11]. Additionally, a small amount of buccal and lingual surface wear was generated by removing a slight layer of these tooth surfaces from the posterior test teeth. This was performed through a slight touch with a laboratory straight handpiece. The removed substance from each tooth surface had a maximum extent of 3 to 5 mm and a vertical depth of approximately 100–250 μm (Appendix A). Two intact teeth, bilaterally distributed besides each grinded tooth, along with additional unaltered adjacent anatomical structures, comprised the superimposition reference areas to obtain the gold standard (true) value. These areas were identical in the before and after tooth wear simulation models, and thus, a perfect registration of the two models was possible, allowing for the determination of the true amount of occlusal tooth wear [10,11,13,17].

### 2.3. 3D Model Acquisition

A reliable 3D surface scanner (accuracy < 20 μm; Laboratory scanner D104a, Cendres+Métaux SA, Biel/Bienne, Switzerland) was used to scan the dental casts prior (T0) and following the tooth wear simulation process (T1). Each entire dental model consisted of 600.000–900.000 triangles, with a premolar clinical crown consisting of approximately 17.000 triangles. Subsequently, the digitized 3D Standard Tessellation Language (STL) models were imported for further processing and analysis in Viewbox 4 software (version 4.1.0.1 BETA, dHAL Software, Kifissia, Greece, http://www.dhal.com/download.htm, accessed on 15 March 2021).

### 2.4. Tooth Wear Measurement

To assess the volume of occlusal tooth wear, the grinded tooth crowns (T1) were selected through manual segmentation and were compared with the original crowns (T0). This process was performed applying the gold standard and five test techniques.

To obtain the gold standard (true) measurements (GS), the segmented T0 and T1 crown models were superimposed on the intact adjacent teeth and alveolar processes (Figure 1a). Thus, the perfect congruence of the two models in these areas after the application of the best-fit algorithm, ensured the accurate 3D occlusal tooth wear assessment [10,11,13,17].

Parts of the buccal and palatal T0 tooth crown surfaces were used as superimposition references for the first group of measurements (PC: partial crown) (Figure 1b). The buccal and lingual surfaces were used, irrespective of buccolingual wear for two reasons. First, to simulate real life conditions, where buccolingual tooth wear is usually undetectable with natural vision. Secondly, in cases where an extended buccal or lingual surface area would have been affected, there would be no other alternative. The second group of measurements was performed using the whole T0 clinical crown as superimposition reference (CC: complete crown; Figure 1c).

Corresponding T0 and T1 3D models were superimposed on each reference area using different settings of an iterative closest point algorithm (ICP) [18]. A manual approximation of the two objects always preceded the automatic algorithm application, to facilitate the process. Each registration was performed, always starting from the original initial position. For each superimposition session, the registration process was applied repeatedly (usually four to five times), until the minimum possible distance between the superimposed models was reached.

Firstly, the outcomes of four different ICP settings were assessed. Setting (A) consisted of 100% estimated overlap of meshes, exclude overhangs, matching point to plane, exact nearest neighbor search, 100% point sampling, and 50 iterations. Setting (B) was defined at 80% estimated overlap of meshes, with all other parameters set as in setting (A). In setting (C), the operator defined freely, for each individual measurement, the estimated overlap of meshes, in an iteratively improved superimposition process regarding the superimposed teeth and the adjacent intact structures. All other parameters were set as in (A). The value used for each case to provide the best possible overlap, was noted in an Excel sheet (Microsoft Excel, Microsoft ©, Richmond, WA, USA). A fourth setting (D), which was otherwise same as (A), was defined in 30% estimated overlap of meshes, based on the average of these values (average: 34.0, SD: 9.3, range: 15–50% estimated overlap of meshes). The specific value was chosen because further assessment of individual cases showed that the lower values provided satisfactory results in all cases and approximated better the gold standard measurements than the higher values [10,11]. Table 1 provides an overview of all techniques assessed in this study (combinations of ICP settings and reference areas).

Based on a previously published protocol [10,11], after each superimposition, the acquired T0/T1 3D tooth crown models were simultaneously sliced using one (gingival) to three slicing planes (gingival, mesial, and distal). The number of the required planes was defined by the need to slice each crown through straight lines, without leaving any uneven margins on the sliced surface model. In advance of the slicing, the planes were positioned with the help of a color map in such way that the occlusal tooth wear surface was included, but the buccolingual tooth wear surfaces were excluded or kept to a minimum. Afterwards, a specific hole filling process was applied on both T0 and T1 occlusal tooth parts, aiming to fill them identically (Figure 2). Certain cases required more than one slicing plane, to ensure identical filling of both models. There, the contralateral points of sharp edges had to be connected, to split the hole of each crown model resulting from slicing. Consequently, the edges of each pair of T0/T1 holes lied on a single plane, allowing for a unique hole closure each time (irrespective of the software’s algorithm), through a flat surface. Thus, the subsequent T0 and T1 watertight 3D models of the superimposed teeth differed only occlusally, being otherwise identical (Figure 3). The volumetric difference of the resulting two models provided the occlusal wear amount of each tooth.

The tooth wear amount detected through the gold standard technique (volume loss of tooth structure in mm^3^) was then compared to that of the test techniques.

A thorough assessment of intra-and inter-operator of similar techniques were published previously [10]. Here, we tested the intra-operator error of the gold standard technique and the technique of choice by repeating the whole process for 10 randomly selected teeth, 1-month after the first set of measurements.

### 2.5. Statistical Analysis

The present statistical approach, performed using IBM SPSS statistics for Windows (Version 25.0, IBM Corp, Armonk, NY, USA), is identical to a previously published by our group in a similar study [10]. Certain deviations from normality (Kolmogorov–Smirnov and Shapiro–Wilk tests) designated the use of non-parametric statistics.

Box plots were used to show the agreement of different techniques with the gold standard technique (trueness) in tooth wear measurement. Any deviation from zero indicates reduced trueness. For each technique, individual value variation from the median indicates precision. Friedman’s test was used to test differences in trueness and precision among different techniques. This was followed by Wilcoxon’s signed rank test for pairwise comparisons, when significant results were evident.

Within each technique, visual inspection of relevant plots and unpaired comparative tests were performed to assess potential effects of crowding status, tooth type, or tooth wear amount on trueness and precision.

Bland Altman plots of tooth wear measurements were created to assess intra-operator error. Deviations from zero indicate reduced precision. The Mann–Whitney U test was applied to assess differences in the reproducibility between techniques.

In all cases, a two-sided significance test was carried out at an alpha level of 0.05. In case of multiple comparisons, a Bonferroni adjustment was applied to the level of significance to reduce the probability of false positive results.

## 3. Results

There was no systematic error between the repeated measurements performed through the gold standard technique and the CC(D) technique of choice (one sample *t*-test, *p* > 0.05). Tooth type or amount of tooth wear did not seem to affect reproducibility. The gold standard measurement and technique CC(D) showed sufficient and comparable reproducibility, overall (*p* > 0.05) and in individual measurements (max difference < 0.040 mm^3^) (Appendix A).

All tested techniques differed significantly to each other in their trueness, except for PC(C) with all CC techniques (Friedman test: *p* < 0.001; Wilcoxon signed rank test: *p* < 0.001; Figure 4). There were differences in precision among techniques (Figure 4). The CC(D) technique provided accurate tooth wear assessment, since it consistently showed excellent agreement with the GS technique (median difference: 0.045, max: 0.219 mm^3^).

Tooth type affected the trueness and precision only of CC(B) technique (Mann–Whitney U test, *p* < 0.01), with molars showing slightly higher differences to the GS measurements (Figure 5a). Tooth alignment in the dental arches affected only the results of the PC(C) technique (Mann–Whitney U test: *p* < 0.01). Tooth wear amount affected significantly only the results of the CC(B) and CC(C) techniques (Kruskal–Wallis test, *p* < 0.01; Figure 5b), with differences from the GS measurements increased with increasing tooth wear amount.

The technique of choice CC(D) did not differ considerably from the GS technique for any tooth type or wear amount (Appendix A). There was a tendency for lower trueness and precision for molars (Mann–Whitney U test, *p* = 0.088) or with increasing amount of tooth wear (Kruskal–Wallis test, *p* = 0.062), but the differences were not statistically significant.

## 4. Discussion

The present study validated a 3D superimposition method for occlusal tooth wear assessment on teeth that undergo limited additional changes in non-occlusal surfaces. The method was previously developed to visualize and quantify occlusal tooth wear on teeth that remain intact in all surfaces, apart from the occlusal [10]. Several modifications in the superimposition reference areas and settings were tested, but the previously defined approach [10] provided the most favorable outcomes, also in the newly tested clinical scenario. Thus, the applicability of the existing highly accurate technique in cases of intact non-occlusal surfaces [10] is expanded to cases that undergo limited additional changes in non-occlusal surfaces. The technique is applicable under regular clinical conditions, since it uses mainstream software applications and 3D digital dental models, which can be easily obtained through intraoral scans [13]. Furthermore, it requires the existence of two intraoral models, obtained at different time points, to monitor tooth or material wear that occurred during this time. For this reason and due to the very high occurrence of tooth wear in the population [19,20,21], it is recommended that a dental scan should be obtained, if possible, in all patients, at the early permanent dentition. This scan, can then be superimposed to scans obtained at later stages. Similarly, immediately after placement of any restoration that needs to be monitored, an intraoral dental model should be obtained. The time span between consecutive diagnostic scans should be defined by the dentist on an individual basis, depending on the needs and the risk factors of each case. Conventional impressions and subsequent stone models can be scanned afterwards, and might also serve this purpose, but apart from material costs and storage needs, they can also be subjected to dimensional changes over time [22].

In a previous report [10], we validated this method for teeth that were not affected in any other surface than the occlusal. However, alterations of the buccal or palatal tooth surfaces can also occur during function. These can be due to bacteria, erosion, or dental interventions, including fillings, tooth whitening, or removal of orthodontic bracket bonding agents. Additionally, the scanner error itself can be a source of such morphological differences [13,17]. A recent study on surface-based 3D superimposition in palatal areas [17] showed that even small artifacts might significantly affect superimposition outcomes. Thus, our previously suggested technique might have not been suitable for teeth subjected to slight morphological changes in non-occlusal surfaces. Following thorough testing of various modifications, it became evident that the previous technique [10] also performed favorably in these cases. Thus, this technique is suitable for occlusal wear assessment in cases that have either no or limited changes in non-occlusal surfaces. Providing that serial crowns are adequately and similarly superimposed in both the above mentioned cases, we also suggest the technique as suitable to assess buccal or lingual tooth wear, at least of a similar amount and pattern to that tested here. In individual patients, such occurrences can be verified following visualization of color coded distance maps of superimposed crowns. Lack of congruence of the two models indicates worn surfaces. It should be noted here that, according to another previous study by our group, certain modifications of the applied 3D superimposition settings should be made in case of extensive changes on non-occlusal tooth surfaces, additional to occlusal wear [11].

The accuracy of the superimposed tooth surface models directly affects the trueness of the wear assessment. Previous studies have shown that the accuracy of an intraoral scan is affected by several factors, such as the scanner, the type, extent, and irregularity of the scanned surface, the software used to process the images and create the final 3D model, and others [13,23,24,25]. The dental surface models of the present study were acquired using a high accuracy laboratory scanner. With this scanner, the distance between superimposed corresponding surfaces, obtained from repeatedly scanned single jaw models, is consistently smaller than 5 μm [11]. For small structures, such as single natural teeth, this scanner is comparable to the current high-end intraoral scanners [13]. Thus, the clinical applicability of the suggested technique, which is applied in individual tooth crown models, is not questioned for natural teeth. However, in cases of restored teeth, the tooth model to be superimposed might show reduced trueness. In such cases, if the model error is comparable to the simulated buccolingual surface wear in this study, the accuracy of the suggested method is expected to be similar to that detected here. In contrast, in cases of larger error, the superimposition outcomes might be affected considerably. The laboratory scanner was used to assure negligible effects of scanner inaccuracy on the gold standard outcome, obtained using a more extended superimposition reference area. According to the study design, this area was not changed during tooth wear simulation, and thus, identical surfaces were available in both pre- and post-wear models. These identical surfaces were perfectly superimposed, and thus, revealed the true crown morphology changes due to tooth wear. This is only feasible under the present experimental conditions, but it provides the true wear values to be compared with those of the single crown techniques that were tested here and are clinically feasible. The matching software used in the present study shows a difference smaller than 0.001 μm between identical duplicates of a surface model, superimposed with the methodology used here [11].

In actual clinical conditions, the gingival margin area may also change over time [15], leading to differences in the adjacent clinical crown surfaces, captured by the scanner each time. This does not affect the performance of the technique if, as suggested, the “exclude overhangs” option is selected. With the proposed settings, the software ignores any surfaces that exceed the outer limits of one surface model over the other, when applying the best fit algorithm. This issue was extensively tested previously, by selecting small reference areas only in one model and superimposing them to much larger models, and it always worked properly [10,11,13,17,26,27,28]. Therefore, it is suggested that, in actual clinical data with large differences in clinical crown sizes over time, the smaller clinical crown should be selected as a superimposition reference. Depending on the time between subsequent models and on individual case characteristics, this might be the initial model, in cases of severe gingival recession development, or the final model, in cases of severe tooth wear.

Apart from proper registration, accurate volume measurements presuppose proper hole filling of the crown parts that are compared. The hole filling process used to create watertight models—a prerequisite for volume measurement—was tested previously [10] and consistently provided identical volumes. The level of crown slicing is also important considering the reproducibility of the measurements, which proved to be high in our study. However, the slicing process does not affect the accuracy of the technique, since following a valid superimposition of serial tooth crown models, the operator can define the type of measurement (point/area distance or volume) and the area where differences need to be measured. A problem might arise in studies including patients with extensive tooth wear in multiple surfaces. In this case, the consistent selection of an anatomically defined area of interest or the consistent slicing at the same anatomical level among various individuals, in order to obtain comparable measurements, might be more challenging.

Tooth wear simulation was performed at three different levels, representing approximately 0.5 to 2 mm vertical loss. This approach aimed to test varying degrees of wear severity, in the presence of clinically significant tooth structure loss. The high performance of the technique was consistent, independent of the different tooth wear amounts or tooth types tested. This was also evident in previous studies [10,11]. Smaller amounts of occlusal tooth wear were not simulated here, though they might be present in actual patients. We decided to test this range, since the performance of the ICP algorithm would be equally good, if not better, with smaller alterations than the one tested here (Appendix A) [11]. It is important to note that the suggested technique also performs perfectly in cases where the serial tooth crown models are identical (Appendix A) [13,17,26]. Existing clinical studies, as well as clinical experience, suggest that, taken together, the above scenarios might cover the majority of cases to be managed in a conventional dental practice [29,30]. Although quantitative data regarding the prevalence, type, and amount of wear in buccal and lingual tooth surfaces are scarce, the same seems to apply for the limited buccal and lingual surface changes simulated in this study [14]. 

So far, several methods were suggested in the literature for tooth wear monitoring [7,31,32,33]. Qualitative methods usually suffer from reduced precision. On the other hand, the main shortcoming of most previous quantitative methods is that they were not tested against a true measurement [7,34]. Most previous studies focused on reproducibility and some based their assessments only on differences between group means. The assessment of differences in means is inadequate when testing the performance of methods that should work properly in every single case [35]. Furthermore, reproducibility, even if it is high, does not guarantee the trueness of a measurement. In agreement with previous studies [10,11,17], the superimposition of serial models on intact structures accomplished perfect registration in the present study, and thus, provided the true measurement. This would not be possible in an in vivo design, where no neighboring structure could be considered intact. Thus, through this design, we developed a clinically applicable superimposition technique that can provide a measurement quite similar to the true. Furthermore, differences were assessed between group means, but also for individual cases, testing both precision and trueness.

There are certain quantitative techniques with comparable accuracy to that of the present method, but they require complex, expensive equipment, and expertise to be properly applied [7]. This limits the applicability of these techniques. On the contrary, we present a technique that can be incorporated in everyday clinical practice, provided that specific software is available and short training is performed. Following this, the application of the method to measure the wear of one tooth requires approximately 2–3 min. In case of time limitations, such as in large-scale studies or in a busy practice, the use of index teeth might be a viable solution, as suggested recently [30]. Furthermore, most previous techniques assessed vertical loss of tooth structure [7]. However, this information is quite limited compared to the assessment of tooth loss in the original whole crown 3D space. With the suggested method, results can be visualized through color-coded distance maps and quantified as desired by the operator. For example, the vertical loss at any point, the mean vertical loss of a selected surface area, or any volume loss can be easily measured, following proper registration of two or more serial tooth models. A freeware and easy to use software (WearCompare; leeds-digitaldentistry.com; accessed on 22 March 2021) [36] for serial tooth surface model superimpositions to assess occlusal tooth wear was also recently suggested. This performed perfectly when duplicated surfaces were superimposed, but relatively large errors became evident when the original position of the duplicated models changed or when actual patient data were considered [36]. Furthermore, its robustness in the presence of changes in non-occlusal surfaces that are used as superimposition references has not been tested yet against a gold standard. A recent study testing the agreement of this method to a qualitative one (BEWE index) reported limited sensitivity and specificity [37]. Nevertheless, the latter finding should be interpreted with caution, since the qualitative approaches do not allow for an accurate wear quantification [5,6,38]. Thus, their use as a reference to test highly accurate methods, as the one reported here, might not be appropriate.

The present study implemented an in vitro design, which allowed the acquisition of the gold standard measurement (true value) to be compared with the outcomes of other clinically applicable methods. Currently, there is no reliable way to obtain the true wear value using intraoral scanners in vivo. Although this might be a limitation of the study, for reasons discussed in detail in the manuscript and are based on current evidence, we believe that the present design allows for direct transfer of our findings to actual conditions. The study focused on posterior teeth since these are round-shaped and this might restrain the performance of the best-fit algorithm [10], in contrast to the more rectangular anterior teeth. The performance of similar techniques on anterior teeth was thoroughly tested previously and proved highly reproducible [10,11,13,17,26]. Thus, if the present method is valid for posterior teeth, it is expected to work equally good, if not better, in the anterior dentition [10,11,17,26]. Reproducibility was tested for the gold standard and the technique of choice. The other techniques were not tested, since they showed reduced trueness. Thus, even if they were highly reproducible [10], this does not assure that they would provide valid assessments. Furthermore, the reproducibility of similar techniques was thoroughly tested previously and consistently provided satisfactory results [13,17,26]. Finally, this study tested the performance of 3D superimposition techniques to assess occlusal tooth wear in presence of limited changes in non-occlusal surfaces. When larger changes occur, such as when a bonded wire retainer is placed during the assessment period [16] or when severe tooth wear occurs in the buccal or lingual tooth surface [14,21], specific modifications should be applied [11].

## 5. Conclusions

The present in vitro study tested the trueness and precision of various tooth crown superimposition techniques, used to assess occlusal wear in presence of limited morphological changes on non-occlusal surfaces. The results indicate that the high performance of a previously suggested technique, for teeth with isolated occlusal surface changes, is also robust in this case. Furthermore, the technique is also suitable to assess non-occlusal wear, at least of limited extent.

The present technique offers a highly accurate 3D tooth wear assessment and can be applicable in a clinical environment, provided that easily acquired skills and usually already available equipment is required.

## Figures and Tables

**Figure 1 diagnostics-11-01033-f001:**
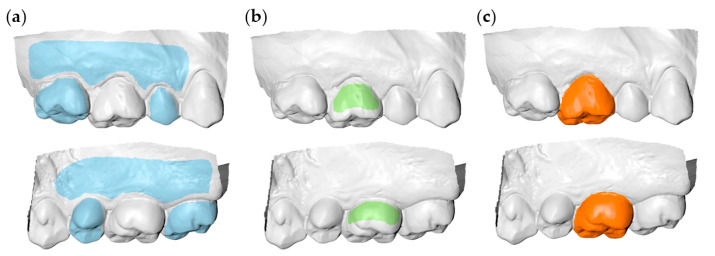
Superimposition reference areas for tooth wear assessment at a maxillary first permanent molar. (**a**) Gold standard area (GS, blue). (**b**) Partial crown area (PC, green). (**c**) Complete crown area (CC, orange). Buccal aspects are shown at the upper row and palatal aspects at the lower row.

**Figure 2 diagnostics-11-01033-f002:**
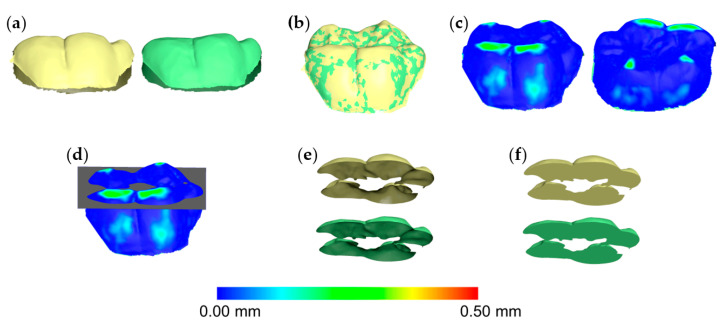
Workflow of tooth wear assessment in a mandibular first molar. (**a**) Tooth crown before (yellow) and after (green) wear simulation at the occlusal and the buccolingual surfaces. (**b**) Tooth crowns superimposed through the complete crown technique with setting D (30% estimated overlap). (**c**) Tooth wear from the buccal (left) and the lingual (right) aspect shown using a color coded distance map. (**d**) Simultaneous slicing of the two crowns using one level (grey). (**e**) Sliced tooth crown parts. (**f**) Watertight models created after the hole-filling process, used to calculate volumes.

**Figure 3 diagnostics-11-01033-f003:**
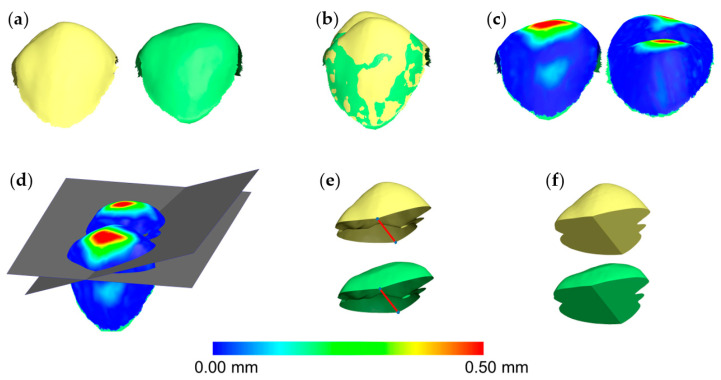
Workflow of tooth wear assessment in a maxillary first premolar. (**a**) Tooth crown before (yellow) and after (green) wear simulation. (**b**) Tooth crowns superimposed through the complete crown technique with setting D (30% estimated overlap). (**c**) Color coded distance map showing the tooth wear from the buccal (left) and the lingual (right) side. (**d**) Two levels (grey) used to simultaneously slice the two crowns. (**e**) Sliced tooth crowns with connected contralateral points (blue) in sharp edges (red line), splitting the hole in two parts to ensure identical hole-filling process. (**f**) Holes filled to create watertight models and, thus, calculate volumes.

**Figure 4 diagnostics-11-01033-f004:**
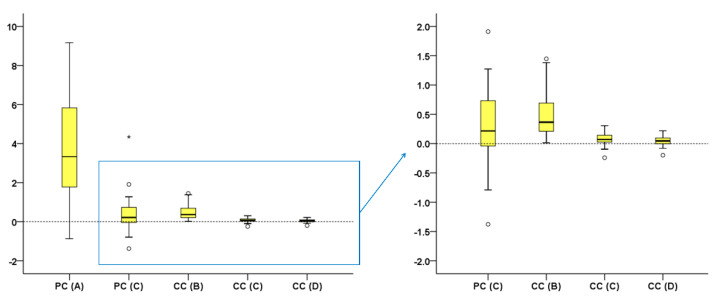
Difference of each technique from the gold standard technique in tooth wear assessment shown in box plots (*y*-axis, mm^3^). The upper limit of the black line represents the maximum value, the lower limit the minimum, the box the interquartile range, and the horizontal black line the median value (trueness). Outliers are shown as black circles (°) or asterisks (*), in more extreme cases, with a step of 1.5 × IQR (interquartile range). Zero value (dashed horizontal line) indicates perfect agreement with the gold standard. Precision is indicated by the vertical length of each plot. The blue box on the right image indicates the area of the graph shown in the left image in a larger scale.

**Figure 5 diagnostics-11-01033-f005:**
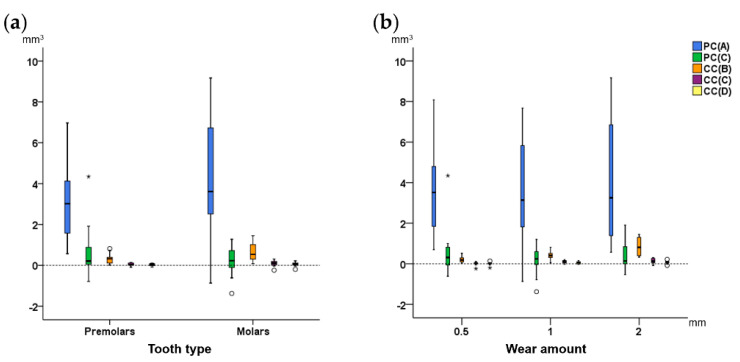
Difference of each technique with the gold standard technique in tooth wear measurements, (**a**) by tooth type, and (**b**) by amount of tooth wear, shown in box plots. The upper limit of the black line represents the maximum value, the lower limit the minimum, the box the interquartile range, and the horizontal black line the median value (trueness). Outliers are shown as black circles (°) or asterisks (*), in more extreme cases, with a step of 1.5 × IQR (interquartile range). Zero value (dashed horizontal line) indicates perfect agreement with the gold standard. The vertical length of each plot indicates precision.

**Table 1 diagnostics-11-01033-t001:** Superimposition techniques tested in the study.

Technique	Reference Area	Estimated Overlap
GS	Adjacent intact teeth and alveolar processes	100%
PC(A)	Buccolingual surfaces	100%
PC(C)	Buccolingual surfaces	User defined
CC(B)	Complete crown	80%
CC(C)	Complete crown	User defined
CC(D)	Complete crown	30%

## Data Availability

All data presented in this study are available in the article and the Appendix A. The material used to generate the data is available upon reasonable request from the corresponding author.

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
