# Peer review of "3D Occlusal Tooth Wear Assessment in Presence of Limited Changes in Non-Occlusal Surfaces"

_diagnostics, 2021, doi:10.3390/diagnostics11061033_

Round 1
Reviewer 1 Report
This study investigated the use of a previously published technique that relies on different approaches to utilise iterative closest point algorithm to monitor tooth wear progression. The manuscript presentation of the technique is clear with several figures provided. The authors compare and contrast their proposed technique with previously published ones and identify its advantages and limitations. Other than minor presentation issues (such as the opening paragraph in the discussion which needs to be sectioned into shorter sentences), I do not have any reservations. Well done.
Reviewer 2 Report
The title is misleading, and should be rephrased.
In Introduction section, the authors should provide informations concerning the main used teeth wear assessment techniques/grading, their flaws and the significance of imaging techniques.
The purpose of the study is unclearly written − if the authors want to assess only the occlusal wear either both occlusal and non-occlusal surfaces.
In material and method 2.2 section - the bilateral non-worn teeth were the same as the studied ones (molar/premolar) or were they random? which was the support of the removed substance both in surface and depth?
For the Discussion section - Does the scanner′s accuracy and results are affected by teeth presenting multiple structures (enamel, metal, ceramic)? Does the presented method can be adjusted to dental restoration materials?
The authors should discuss the factors which influence the obtained wear values and techniques′ accuracy.
Which is the surface and depth wear assessment limitation?
Round 2
Reviewer 2 Report
The paper has been improved after considering the reviewers comments.
This manuscript is a resubmission of an earlier submission. The following is a list of the peer review reports and author responses from that submission.
Round 1
Reviewer 1 Report
Interesting study, but almost same methodology and same results with the previous studies of Ref. 10 ,11.
The authors need to describe clearly on the limitation of their study.
This study was performed using stone model, not under the real clinical situation.
Please delete the first sentence of Discussion. You have previous studies ref. 10 and 11.
Author Response
Response to Reviewer 1
Comment: Interesting study, but almost same methodology and same results with the previous studies of Ref. 10 ,11.
Response: We are glad that the reviewer found the study interesting. Indeed, the methodology is similar to previous studies from our group (Refs 10 and 11), and therefore, it is well tested. However, the present study has a clear aim, which is important and different from the previous studies, expanding the applicability of the suggested methods.
Comment: The authors need to describe clearly on the limitation of their study. This study was performed using stone model, not under the real clinical situation.
Response: The study implemented an in vitro design, allowing the acquisition of the gold standard measurement (true value) to be compared with the outcomes of the other methods that are applicable under real clinical conditions. Currently, the in vitro methodology is the only way to obtain the true value and for this reason we decided to apply this design. However, for reasons discussed in detail in the manuscript and supported by research findings by our and other groups, we believe that the present design allows for direct transfer of our findings to actual clinical conditions. This is now clearly reported in the limitations of our study, at the end of the Discussion section (lines 410-416).
Comment: Please delete the first sentence of Discussion. You have previous studies ref. 10 and 11.
Response: The first sentence of the discussion refers to “…teeth that underwent limited additional changes in non-occlusal surfaces”. This is indeed the first study that tests this scenario, which is often clinically present. Thus, we prefer to report this accordingly.

Reviewer 2 Report
This was a very interesting study attempting to investigate the accuracy of various settings for quantifying tooth wear using surface-matching software. The manuscript, however, has a number of issues that need to be addressed/clarified:
- More comprehensive comparison with similar studies that involved full-crown superimposition and deviation analysis: Marro, F., Jacquet, W., Martens, L., Keeling, A., Bartlett, D., & O’Toole, S. (2020). Quantifying increased rates of erosive tooth wear progression in the early permanent dentition. Journal of dentistry, 93, 103282.; Ahmed, K. E., Whitters, J., Ju, X., Pierce, S. G., MacLeod, C. N., & Murray, C. A. (2017). Clinical monitoring of tooth wear progression in patients over a period of one year using CAD/CAM. International Journal of Prosthodontics, 30(2), 153-155.
- A 'perfect' registration is practically non-existent in best-fit (iterative closest point). I understand what the authors mean but I do not think this is an accurate and true descriptor;
- I'm unclear as to how the gold standard superimposition model involves the use of buccal soft-tissues. How can this be reliable clinically? The authors touched base on this in the discussion but needs expanding. Please include the number of triangles/cloud points involved in the ICP. The description of the superimposition;
- Figure 1 - this looks like a maxillary arch - please revise 'lingual' to palatal in caption;
- Figure 3 is especially helpful to understand the workflow.
Discussion
- I do not think that the opening statement is accurate, consider rephrasing; other clinical studies have attempted to clinically quantify tooth wear that was not necessarily confined to occlusal/incisal surfaces: O’Toole, S., Lau, J. S., Rees, M., Warburton, F., Loomans, B., & Bartlett, D. (2020). Quantitative tooth wear analysis of index teeth compared to complete dentition. Journal of dentistry, 97, 103342. Mehta, S. B., Bronkhorst, E. M., Crins, L., DNJ Huysmans, M. C., Wetselaar, P. P., & Loomans, B. A. (2020). A comparative evaluation between the reliability of gypsum casts and digital grayscale intra‐oral scans for the scoring of tooth wear using the Tooth Wear Evaluation System (TWES). Journal of oral rehabilitation.
- I think the authors need to further analyse the significance of their findings and their clinical applicability;
- The authors do somewhat discuss the limitations of their approach, however, this needs further expanding. The reliance on stone casts that undergo expansion might pose a major limitation: Ahmed, K. E., Whitters, J., Ju, X., Pierce, S. G., MacLeod, C. N., & Murray, C. A. (2016). A proposed methodology to assess the accuracy of 3d scanners and casts and monitor tooth wear progression in patients. Int J Prosthodont, 29(5), 514-521. The use of splicing using planes then filling the surfaces is ideal for wear affecting specific surfaces. If, however, wear impacts multiple surfaces including the cervical (NCCLs) then splicing might be challenging. Finally, the workflow, while accurate and similar to previous studies, remains labour and time intensive, requiring specialised software and training. Can it be used in large scale studies? How would it compare to other software such as wearcompare or clinical monitoring? Maybe the use of index teeth might help? O’Toole, S., Lau, J. S., Rees, M., Warburton, F., Loomans, B., & Bartlett, D. (2020). Quantitative tooth wear analysis of index teeth compared to complete dentition. Journal of dentistry, 97, 103342.
Author Response
Response to Reviewer 2
Comment: This was a very interesting study attempting to investigate the accuracy of various settings for quantifying tooth wear using surface-matching software. The manuscript, however, has a number of issues that need to be addressed/clarified:
Response: We are happy to see that the reviewer found our study interesting and we did our best to improve it based on the reviewers’ comments.
Comment: More comprehensive comparison with similar studies that involved full-crown superimposition and deviation analysis: Marro, F., Jacquet, W., Martens, L., Keeling, A., Bartlett, D., & O’Toole, S. (2020). Quantifying increased rates of erosive tooth wear progression in the early permanent dentition. Journal of dentistry, 93, 103282.; Ahmed, K. E., Whitters, J., Ju, X., Pierce, S. G., MacLeod, C. N., & Murray, C. A. (2017). Clinical monitoring of tooth wear progression in patients over a period of one year using CAD/CAM. International Journal of Prosthodontics, 30(2), 153-155.
Response: Thank you for the suggestions. These references have been added in the manuscript along with relevant discussion regarding the first one (lines 407-409). The second has been also added where applicable (lines 290-292), although it did not report any method error assessment.
Comment: A 'perfect' registration is practically non-existent in best-fit (iterative closest point). I understand what the authors mean but I do not think this is an accurate and true descriptor;
Response: A perfect registration at a level of at least 0.0001 mm mean absolute distance between superimposed models is possible, for example, when identical models are superimposed. The deviation from 0 is just a software limitation, but such a distance can be considered 0 for our purpose. Therefore, we would prefer to retain this reporting, understanding the concern of the reviewer.
Comment: I'm unclear as to how the gold standard superimposition model involves the use of buccal soft-tissues. How can this be reliable clinically? The authors touched base on this in the discussion but needs expanding.
Response: The study implemented an in vitro design, allowing the acquisition of the gold standard measurement (true value) to be compared with the outcomes of the other methods that are applicable under real clinical conditions. The buccal soft tissues were used only for the gold standard measurement, which is not clinically applicable. Currently, the in vitro methodology is the only way to obtain the true value and for this reason we decided to apply this design. However, for reasons discussed in detail in the manuscript and supported by research findings by our and other groups, we believe that the present design allows for direct transfer of our findings to actual clinical conditions, using of course the clinically applicable method suggested. This is now clearly reported in the limitations of our study, at the end of the Discussion section (lines 410-416).
Comment: Please include the number of triangles/cloud points involved in the ICP. The description of the superimposition.
Response: We thank the reviewer for noticing this. This information has now been added in the manuscript for the entire dental models used and for a single tooth crown (lines 102-104). These triangles are more or less equally distributed over the models and describe the anatomical morphology with quite high resolution. Therefore, the exact measurement of triangles/vertices for different smaller areas can be inferred, although not evaluated directly for the whole sample. We do not consider that further detailed information on specific number of triangles per case would be meaningful.
Comment: Figure 1 - this looks like a maxillary arch - please revise 'lingual' to palatal in caption.
Response: Revised as suggested.
Comment: Figure 3 is especially helpful to understand the workflow.
Response: We are happy to read this comment.
Discussion
Comment: I do not think that the opening statement is accurate, consider rephrasing; other clinical studies have attempted to clinically quantify tooth wear that was not necessarily confined to occlusal/incisal surfaces: O’Toole, S., Lau, J. S., Rees, M., Warburton, F., Loomans, B., & Bartlett, D. (2020). Quantitative tooth wear analysis of index teeth compared to complete dentition. Journal of dentistry, 97, 103342. Mehta, S. B., Bronkhorst, E. M., Crins, L., DNJ Huysmans, M. C., Wetselaar, P. P., & Loomans, B. A. (2020). A comparative evaluation between the reliability of gypsum casts and digital grayscale intra‐oral scans for the scoring of tooth wear using the Tooth Wear Evaluation System (TWES). Journal of oral rehabilitation.
Response: We agree with the reviewer that there are various other studies assessing tooth wear in non-occlusal surfaces and some of them are also reported in our paper. However, to our knowledge there is no study testing any such method against a gold standard reference (true value). This is exactly what the first sentence of the introduction states. We would like to keep this phrase to emphasize the originality of the present study, but also to underline the need for further research in the field. Reproducibility is necessary for a method to be considered valid, but trueness is also fundamental. From the two studies mentioned in this comment, the first tests intra- and inter-operator agreement, but not trueness. The second study tests an index and lacks quantification of tooth wear amount. Thus, they do not contradict our statement.
Comment: I think the authors need to further analyse the significance of their findings and their clinical applicability; The authors do somewhat discuss the limitations of their approach, however, this needs further expanding. The reliance on stone casts that undergo expansion might pose a major limitation: Ahmed, K. E., Whitters, J., Ju, X., Pierce, S. G., MacLeod, C. N., & Murray, C. A. (2016).. Int J Prosthodont, 29(5), 514-521.
Response: We thank the reviewer for pointing several interesting issues that worth to be discussed. Regarding the specific comment, actually, we present a digital 3D superimposition method that can be applied in any digital model, either directly or indirectly obtained. We just used stone models for the needs of the present in vitro simulation study. We revised the first paragraph of the discussion to clarify this issue, and we added a relevant comment and the suggested reference at the end of this paragraph.
Comment: The use of splicing using planes then filling the surfaces is ideal for wear affecting specific surfaces. If, however, wear impacts multiple surfaces including the cervical (NCCLs) then slicing might be challenging.
Response: As reported previously in the manuscript, “…the slicing process does not affect the accuracy of the technique, since following a valid superimposition of serial tooth crown models, the operator can define the type of measurement (point/area distance or volume) and the area where differences need to be measured.”. However, we agree with the reviewer that in certain cases the definition of the slicing level or the area of interest is important. To underline this, we added the following text in the relevant part of the discussion (lines 346-349): “A problem might arise in studies including patients with extensive tooth wear in multiple surfaces. In this case, the consistent selection of an anatomically defined area of interest or the consistent slicing at the same anatomical level among various individuals, in order to obtain comparable measurements, might be more challenging.”
Comment: Finally, the workflow, while accurate and similar to previous studies, remains labour and time intensive, requiring specialised software and training. Can it be used in large scale studies? How would it compare to other software such as wearcompare or clinical monitoring? Maybe the use of index teeth might help? O’Toole, S., Lau, J. S., Rees, M., Warburton, F., Loomans, B., & Bartlett, D. (2020). Quantitative tooth wear analysis of index teeth compared to complete dentition. Journal of dentistry, 97, 103342.
Response: Relevant information, including the suggested reference, is now provided in the manuscript in Lines 390-394: “…we present a technique that can be incorporated in everyday clinical practice, provided that specific software is available and short training is performed. Following this, the application of the method to measure the wear of one tooth requires approximately 2-3 minutes. In case of time limitations, such as in large scale studies or in a busy practice, the use of index teeth might be a viable solution, as suggested recently [31].”. Furthermore, the software that we use is one of the available software in the market and is relatively low cost. However, we would not like to expand more on this within manuscript, since this might be considered advertising of the software, which is something that we do not wish.

Round 2
Reviewer 2 Report
The authors have made a clear effort to address the forwarded comments, introducing certain changes to their manuscript, and providing a cogent response to many of points raised, significantly improving the quality of the submission. Whilst I still have some reservations on the use of certain terms, and the novelty and clinical applicability of the proposed methodology, yet, these reservations do not detract from the value of the study and would fall under reasonable academic debate. In my opinion, this manuscript is now ready for typesetting.